# Identification of New Chromosomal Loci Involved in *com* Genes Expression and Natural Transformation in the Actinobacterial Model Organism *Micrococcus luteus*

**DOI:** 10.3390/genes12091307

**Published:** 2021-08-25

**Authors:** Enzo Joaquin Torasso Kasem, Angel Angelov, Elisa Werner, Antoni Lichev, Sonja Vanderhaeghen, Wolfgang Liebl

**Affiliations:** 1Chair of Microbiology, Technical University of Munich, Emil-Ramann-Str. 4, 85354 Freising, Germany; ejtorasso20@gmail.com (E.J.T.K.); angelov.tum@gmail.com (A.A.); elisawerner@gmx.de (E.W.); tony_lichev@abv.bg (A.L.); s.vanderhaeghen@tum.de (S.V.); 2Institute of Medical Microbiology and Hygiene, University Clinic Elfriede-Aulhorn-Str. 6, 72076 Tübingen, Germany

**Keywords:** competence, natural transformation, Actinobacteria, ComEA, ComEC, *tad*-cluster

## Abstract

Historically, *Micrococcus luteus* was one of the first organisms used to study natural transformation, one of the main routes of horizontal gene transfer among prokaryotes. However, little is known about the molecular basis of competence development in *M. luteus* or any other representative of the phylum of high-GC Gram-positive bacteria (Actinobacteria), while this means of genetic exchange has been studied in great detail in Gram-negative and low-GC Gram-positive bacteria (Firmicutes). In order to identify new genetic elements involved in regulation of the *comEA*-*comEC* competence operon in *M. luteus*, we conducted random chemical mutagenesis of a reporter strain expressing *lacZ* under the control of the *comEA*-*comEC* promoter, followed by the screening of dysregulated mutants. Mutants with (i) upregulated *com* promoter under competence-repressing conditions and (ii) mutants with a repressed *com* promoter under competence-inducing conditions were isolated. After genotype and phenotype screening, the genomes of several mutant strains were sequenced. A selection of putative *com*-influencing mutations was reinserted into the genome of the *M. luteus* reporter strain as markerless single-nucleotide mutations to confirm their effect on *com* gene expression. This strategy revealed mutations affecting *com* gene expression at genetic loci different from previously known genes involved in natural transformation. Several of these mutations decreased transformation frequencies by several orders of magnitude, thus indicating significant roles in competence development or DNA acquisition in *M. luteus*. Among the identified loci, there was a new locus containing genes with similarity to genes of the *tad* clusters of *M. luteus* and other bacteria*.*

## 1. Introduction

Horizontal gene transfer (HGT) is a widespread mechanism by which organisms from all three domains of life are able to exchange genetic material [1]. This trait allows them to occupy new ecological niches or to acquire novel virulence factors or antibiotic resistance [2,3]. One of the three main methods for HGT is natural transformation, which has been observed to occur both in bacteria and archaea under conditions that can vary considerably for different transformable species [4,5].

Natural genetic transformation is a parasexual process that requires an extracellular DNA molecule and a competent recipient cell. Since its discovery in 1928 [6], many bacterial species from several taxonomic groups have been shown to bear this trait. Thus far, most of the studies aiming to unveil the molecular mechanism behind natural transformation have been conducted on pathogens such as *Neisseria gonorrhea* [7], *Haemophilus influenzae* [8], *Vibrio cholerae* [9], and *Streptococcus pneumoniae* or model organisms such as *Bacillus subtilis* [10] and *Thermus thermophilus* [11]. While progress has been made regarding the molecular details in Firmicutes (low-GC Gram-positive bacteria) [12,13], almost nothing is known yet for the second major Gram-positive phylum and the Actinobacteria (high-GC Gram-positive bacteria), which are among important pathogens such as *Mycobacterium tuberculosis*. This is remarkable, considering that *Micrococcus luteus* (*M.* “*lysodeiktikus*”) was among the first bacteria used for research on this topic [14,15].

Most bacterial species that use natural transformation rely on similar DNA uptake machineries [5,16]. Certain core genes involved in this process have been shown to be conserved across phylogenetically distant taxa. All transformable bacteria studied thus far harbor the transmembrane channel ComEC and an extra-cytoplasmic DNA-binding protein such as ComEA in *B. subtilis* [4,17], which serves as a receptor to deliver dsDNA to the highly conserved ComEC [18]. The competence (*com*) genes *comEA* and *comEC* on the *M. luteus* genome overlap and are co-transcribed genes, and both have been shown to be essential for transformation [16].

The expression of *comEC*, *comEA*, and other competence genes is often a tightly regulated process that is triggered in certain physiological states or under specific stress conditions. Competence gene expression is mainly activated during the early exponential phase in *S. pneumoniae* and strictly during the stationary phase in *B. subtilis* [19,20]. Although the molecular mechanisms of natural transformation are similar among species, there is a broad variety of regulatory strategies [5]. The best studied competence regulators include alternative σ factors, transcription activators and transcriptional co-regulators. A well-studied alternative σ factor, σ^X^ of *S. pneumoniae*, is encoded by two identical *comX* genes and is widely distributed among the Firmicutes. σ^X^ forms a complex with RNA polymerase that recognizes an eight bp sequence in the promoter of almost all *com* genes that are required for transformation [21], resulting in the induction of competence. Similar mechanisms have been found in other streptococcal species and also in more distantly related phylogenetic lineages within the Firmicutes such as in the Lactobacillales. The most thoroughly studied example of direct transcriptional activation that drives competence development is found in *B. subtilis*. Here, competence development is induced when the ComK protein binds to its own promoter, creating an auto-stimulatory cycle that results in a ComK concentration increase and the development of competence [19,22]. Only 15% of the population becomes competent, resulting in so-called bistability or phenotypic heterogeneity. ComK induces the expression of more than 100 genes, including all the genes required for transformation, and most of these genes are directly regulated by the binding of ComK to recognition sites within their promoters [22,23].

In Gram-negative organisms such as *Haemophilus influenzae*, *Vibrio cholerae*, and others, competence is induced by activation of the required genes through cyclic AMP (cAMP) bound to the receptor protein CRP (cAMP receptor protein). Complex formation of these two molecules is not sufficient to generate competence. In *H. influenzae*, the co-regulator Sxy has been proposed to work as an activating cofactor for CRP, and in *V. cholera*, the Sxy orthologue TfoX presumably works in a similar manner [24,25].

As it has been reported before [16], genes for many of the known proteins involved in DNA uptake and competence acquisition were not found in the *M. luteus* genome, pointing to substantial differences in this organism and maybe in the other members of the actinobacterial phylum. In a recent study, it was proposed that amino acids act as nutritional factors and that the stringent response alarmone (p) ppGpp plays a role in the regulation of natural transformation in *M. luteus* [26]. Our aim in this work was to identify new elements related to the regulation of the co-transcribed genes *comEA* and *comEC* in *M. luteus*.

Considering how certain main elements of the regulation of the late *com* genes have been found in *B. subtilis* and *S. pneumoniae* [27,28,29], we decided to take a similar approach by using ethyl methanesulfonate (EMS) to generate random mutations in the genome of *M. luteus*, followed by a search for mutants with altered *com* gene expression. Similar to *B. subtilis*, *M. luteus* expresses competence (and late *com* genes) under nutritional stress and specifically in a particular physiological state, the late exponential phase [15]. Using a reporter strain that has the *comEA* and *comEC* ORFs replaced by the *Escherichia coli lacZ* gene, we screened for mutants that have high *comEA/EC* promoter activity under repressing conditions (M-mutants) and others that have the promoter completely or partially repressed in competence-inducing minimal medium (R-mutants). After sequencing of selected mutants, we sought to identify mutated regions in the genome that apparently cause the observed phenotypes and thus may have a role in *comEA/EC* promoter regulation. We anticipated that by sequencing of the mutants’ genomes, analysis of the mutations’ distribution and reinsertion of the most interesting ones into the *M. luteus* reporter strain chromosome, it should be possible to identify certain causative mutations and genes that are involved in the regulation of *com* gene expression and natural transformation in this model representative of the Actinobacteria.

## 2. Materials and Methods

### 2.1. Bacterial Strains and Growth Conditions

All strains used in this work are listed in Table 1. All mutant strains are derivatives of the tryptophan auxotroph *Micrococcus luteus* trpE16, which originated from the strain “*Micrococcus lysodeikticus* ISU” [29]. Baffled flasks were used for incubation with either the full medium lysogeny broth (LB) or glutamate minimal medium (GMM) at 30 °C and 180 rpm shaking. LB was prepared according to the Lennox formulation [30], containing Triptone (10 g/L), yeast extract (5 g/L), and NaCl (5 g/L). Glutamate minimal medium (GMM) was prepared according to Wolin and Naylor [31] and consisted of 2 g/L K_2_HPO_4_, 1 g/L NH_4_Cl, 10 g/L sodium glutamate, 7 g/L glucose, 0.1 g/L MgSO_4_, 0.004 g/L FeSO_4_, and 0.002 g/L MnCl_2_. Unless otherwise specified, 0.1 mg/mL of tryptophan was included to enable growth of the Trp auxotrophs. The pH of GMM was adjusted to 7.2 with HCl. When required, plates were prepared by supplementing the media with 1.5% (*w/v*) agar. Strains carrying a kanamycin resistance cassette (*kan*) were cultured in media supplemented with 60 µg/mL kanamycin sulfate. Ampicillin and rifampicin were added at 50 µg/mL and 1 µg/mL as required. For the transformation frequency assays, casein hydrolysate plates (CAH) containing 1% (*w/v*) sodium glutamate, 0.2% (*w/v*) K_2_HPO_4_, 0.1% (*w/v*) NH_4_Cl, 0.01% (*w/v*) MgSO_4_, 0.0004% (*w/v*) FeSO_4_, 0.0002% (*w/v*) MnCl_2_, 0.5% (*w/v*) tryptophan-free acid hydrolyzed casein (EMD Millipore, Danvers, MA, USA), 0.7% (*w/v*) glucose, and 1.3% (*w/v*) agar were used.

### 2.2. EMS Mutagenesis and Mutagenesis Efficiency Assessment

For the introduction of random mutations, cells from an overnight (O/N) full (LB) medium culture of the transcriptional reporter *M. luteus* trpE16 ∆*comEA*/*EC*:*lacZ*-*kan* (Appendix A) [26] were used to inoculate LB with and without supplementation of 5 mM ethyl-methanesulfonate (EMS) and the corresponding antibiotic. Starting at an OD_600nm_ of 0.2, the cultures were incubated for 24 h with shaking at 180 rpm and 30 °C. The treated cells were washed twice with phosphate buffered saline (PBS) by centrifugation. PBS consisted of 137 mM NaCl, 2.7 mM KCl, 10 mM Na_2_HPO_4_, 1.8 mM KH_2_PO_4_, adjusted at pH = 7.4. For cell recovery and to allow segregation of mutations (single nucleotide polymorphisms, SNPs), samples of 0.05 mL were inoculated in 5 mL of fresh LB with kanamycin (LB_Kan_) and incubated with aeration O/N at 30 °C. To assess the efficiency of mutagenesis, mutagen-induced rifampicin resistance was monitored [32]. Samples of EMS-treated cells were plated on full media supplemented with 1 µg/mL of rifampicin, and after growth, the frequency of resistant colonies was determined and compared with the frequency of spontaneous rifampicin-resistant colonies arising from cells not treated with the mutagen. Mutagenized cells were mixed with 25% glycerol (*v/v*; final concentration) and stored at −80 °C until use.

Screenings for two phenotypes were conducted with the mutagen-treated cells: (i) *comEA/EC* promoter-repressing conditions, using LB_Kan_ plates supplemented with 45 µg/mL of 5-bromo-4-chloro-3-indolyl-β-D-galactopyranoside (X-gal), and (ii) *comEA/EC* promoter- and competence-inducing conditions, using GMM containing antibiotic and 80 µg/mL of X-gal (Figure 1A). In this way, two groups of mutants were obtained, the M-mutants showing enhanced *comEA/EC* promoter activity on LB (blue colonies on LB/X-gal plates), a competence-repressing medium [15], and the R-mutants showing no *comEA/EC* promoter activity when grown on GMM minimal medium (white colonies on GMM/X-gal plates) (Figure 1B). Each mutant with dysregulated *comEA/EC* promoter activity was labeled accordingly with either M or R followed by the isolate number.

### 2.3. Linkage Analysis

Since we were looking for mutations in genes involved in the regulation of the *comEA/EC* promoter but at different chromosomal loci than the *comEA/EC* locus itself, mutations occurring within *comEA*/*comEC* or within the *lacZ* gene on the reporter strain’s chromosome had to be identified and excluded from further analysis. Therefore, to evaluate the genetic linkage between the mutations causing each of the phenotypes screened for and the *com* promoter-*lacZ* reporter construct (Appendix A), genomic DNA from each EMS treated mutant encompassing the 7 kb reporter construct and its SNPs was prepared and used to transform naturally competent *M. luteus* trpE16 cells. Only cells transformed with the reporter construct would be able to grow on kanamycin. One hundred microliter aliquots of freshly transformed cell suspensions were plated on the corresponding media supplemented with X-Gal and antibiotic. For a qualitative evaluation of the phenotypes, the M-mutant transformants were plated on LB/X-gal and the R-mutant transformants on GMM/X-gal. Negative controls lacking DNA were always performed with the same cell suspensions. Additional control reactions were prepared, including transformations with genomic DNA from the trpE16 strain as well as from the unmutagenized reporter strain. The linkage degree (%) was calculated dividing the number of colonies with the mutant blue/white phenotype by the ones with the reporter strain phenotype.

### 2.4. β-Galactosidase Activity Assays

Precultures of all the evaluated strains containing the transcriptional reporter Δ*comEA*/*EC*:*lacZ* were grown O/N in LB. Adequate aliquots of the cells were harvested and washed with PBS by centrifugation at 14,000× *g* for 10 min. All samples were normalized such that they would have the same OD_600nm_ of 1.5. All suspensions were treated with 0.1 mg/mL of lysozyme for 10 min at 37 °C, followed by transfer of 200 µL aliquots to a 96-well microtiter plate. For the measurement of LacZ activity, the β-galactosidase substrate 4-methylumbelliferyl β-D-galactopyranoside (MUG) (Sigma-Aldrich, St. Louis, MO, USA) was added to each well at a final concentration of 250 µg/mL. Kinetic measurements were performed by incubating the cells in a FLUOstar Omega (BMG LABTECH, Ortenberg, Germany) microplate reader at 30 °C for 10 h and measuring fluorescence caused by MUG cleavage every 10 min at 460 nm. The wildtype strain of *M. luteus* was incubated with and without a substrate to establish the background fluorescence. There were at least 3 biological replicates and 3 technical replicates for every strain assayed. Controls without cells and without substrate were created to evaluate background noise. To determine the relative activity of the promoter of *comEA*/*EC*, a four parameter logistic function was fitted to the measured kinetic curves. The linear range of the kinetic curves prior to substrate depletion was used for evaluation. This was accomplished by calculating the steepest slope within the linear range of the curves which corresponds to the maximal velocity of the enzymatic reaction and is directly proportional to the enzyme concentration [26].

### 2.5. Genomic DNA Extraction from M. luteus

After O/N incubation in full medium, 4 mL of culture were centrifuged at 14,000× *g* for 5 min. The pellet was resuspended in 25 μL of PBS and mixed with 25 μL of 0.1 mg/mL of lysozyme. A volume of 1 μL of proteinase K was mixed with 300 μL of TC buffer (Biozym Scientific GmbH, Oldendorf), and the mixture was added to the cells. After mixing well, the cells were incubated for 15 min at 65 °C. The suspension was mixed every 5 min. After cooling the sample to 37 °C, 1 μL of 5 μg/μL of RNase A was added and the extract was incubated at the same temperature for 30 min. One hundred fifty microlites of cetyltrimethylammonium bromide (CTAB) was added to the sample after cooling on ice. The mixture was vortexed and centrifuged at 4 °C and 14,000× *g* for 15 min. The supernatant was transferred into a clean 1.5 mL tube and mixed with 500 μ of isopropanol. The suspended DNA was pelleted and washed twice with 70% ethanol. After removing all the ethanol, the DNA pellet was dried and resuspended in 50 μL of ultrapure H_2_O.

### 2.6. Genome Sequencing, Assembly, Quality Analysis, and Mutations Plotting

Genomic DNA was isolated from the *M. luteus* strains and used for sequencing on a MiSeq platform (Illumina, San Diego, CA, USA) with standard manufacturer protocols (TruSeq LT DNA sample preparation kit, 2 × 150 bp paired-end reads). Mapping assembly was performed with the Snippy program (https://github.com/tseemann/snippy (accessed on the 18 August 2021)) [33], using the *M. luteus* trpE16 sequence deposited at the NCBI database (Genbank accession no. CP007437) as a reference.

### 2.7. SNP Clean Insertions and Construction of Gene Deletion Mutants

In this work, all genome modifications were accomplished through natural transformation and homologous recombination. By using the *codBA*-based markerless genome modification system [34], we were able to introduce clean gene deletions and single nucleotide changes in specific sequences of the *M. luteus* chromosome. To this end, pKOS6b genome modification plasmids were first constructed to contain a DNA insert composed of *M. luteus* chromosomal DNA sequences flanking the SNP (or the sequence to be deleted) to be introduced into the chromosome. For SNP introduction, each insert was obtained by amplifying the mutated region from a certain *M. luteus* mutant. All amplicons were a size of 1–2 kbp and were cloned in the linearized pKOS6b vector (6661 bp) through in vitro assembly of equimolar amounts in a 20 µL Gibson Assembly reaction [35]. Competent cells of *E. coli* XL1 were transformed with the Gibson Assembly reaction through heat shock, and transformants were selected on LB plates containing the appropriate antibiotic. After verification of the plasmid by restriction digestion or sequencing, the plasmid was prepared and used to directly transform *M. luteus* competent cells. Uptake of the plasmids resulted in the generation of kanamycin resistant and 5-fluorocytosine (5-FC) sensitive mutants. 5-FC is a non-toxic pyrimidine analogue that is converted by cytosine deaminase (encoded by the *E. coli codA* gene present on pKOS6b) to a toxic compound 5-fluorouracil (5-FU) [36,37]. The gene *codB*, which is located next to *codA* on pKOS6b, codes for a cytosine permease that enhances the penetration of 5-FC into the cell. Prior to counterselection via the *codBA* marker, transformants were first selected on LB plates containing antibiotic (kanamycin). Fifteen of the kanamycin-resistant colonies were selected and cultured in 1 mL LB medium each at 30 °C for 4 h. Cultivation of the transformants under non-selective conditions fosters the survival of cells in which a second homologous recombination and excision of the integrated plasmid occurred. 5-FC resistant strains that arise from plasmid excision can easily be selected on a growth medium containing the analogue. Thus, appropriate dilutions of the cell suspension were plated on LB supplemented with 500 µg/mL 5-FC and incubated at 30 °C for 72h. Recombinant plasmid excision at the locus of interest results either in the desired chromosomal mutation or the wild-type sequence. The distribution of both resulting genotypes is a random process (in theory 50:50) and requires additional tests for confirmation. The deletion of desired genes was verified by locus amplification, and amplicon length comparison to the wild-type. In the case of SNP introduction, the corresponding regions were amplified and sequenced (Appendix A).

### 2.8. Extraction of Total RNA, 5′RACE and 5′UTR Analyses

In order to determine the beginning of the transcripts of interest and better analyze how they are affected, the following techniques were used. For total RNA preparation, 4 mL of cells were sedimented at 14,000× *g* for 5 min and washed with PBS. After resuspending the cells in 1 mL of PBS, 0.1 mg/mL of lysozyme was added. Following 10 min incubation at 37 °C, RNA was isolated with the ZR Fungal/Bacterial RNA Miniprep Kit (Zymo Research, Irvine, CA, USA). An additional rigorous DNase treatment was performed with the TURBO DNA-free Kit (Thermo Fisher Scientific, Waltham, MA, USA) to minimize the amount of genomic DNA left in the samples. Next, the RNA was reverse transcribed into cDNA by using the iScript cDNA synthesis kit (Bio-Rad Laboratories, Hercules, CA, USA) in a 20 µL reaction containing 1 µg RNA. In order to determine the length and sequence of specific mRNAs, the 5′/3′RACE Kit 2nd Generation (Sigma-Aldrich, St. Louis, MO, USA) was utilized. The High Pure PCR product purification Kit (Sigma-Aldrich, St. Louis, MO, USA) was used when required and the final products were sequenced. The sequences of the analyzed genes were subjected to analysis with the widely utilized software tools RNAstructure (https://rna.urmc.rochester.edu/RNAstructureWeb/ (accessed on the 18 August 2021)) and RBS calculator (https://www.denovodna.com/software/predict_rbs_calculator (accessed on the 18 August 2021)) for ribosome binding site (RBS), translational initiation rate, and secondary structures prediction.

### 2.9. Transformation Frequency Assay

For transformation frequency assays, cells from an O/N culture grown in full medium were inoculated into 25 mL GMM (with the respective supplementation if needed) at an initial OD_600nm_ of 0.2. After 20 hours of incubation at 30 °C in a shaking incubator (180 rpm), the OD_600nm_ was determined to serve as a marker for the total cell count, and 2 mL of the GMM cultures were pelleted by centrifugation at 10,000× *g* for 8 min at 4 °C. The supernatant was discarded, and the cells were resuspended in 1 mL transformation buffer (100 mM CaCl_2_, 50 mM Tris-HCl, pH 7.2). Next, 300 ng of plasmid pJET-*trpE*, constructed by cloning of the wild-type *trpE* allele from *M. luteus* ATCC 27141 gene in pJET in *E. coli*, was added to the cell suspension. If internalized by natural transformation, this DNA is suited to allow conversion of trpE16 tryptophan auxotrophic cells back to prototrophy. Following incubation for 30 min at 30 °C in a shaking incubator (180 rpm), the transformation reaction was stopped by placing the cells on ice. The cell suspensions were sonicated using an UP200S sonotrode (Hielscher Ultrasonics GmbH, Teltow, Germany) for 1 min with 30% amplitude and 0.25 duty cycle to gently disrupt cell aggregates before plating on CAH^-^ agar plates to score transformants and on LB plates to determine the total viable cell counts. Control reactions performed without the addition of DNA delivered the rate of spontaneous reversion to prototrophy, which determined the detection limit of the assay.

## 3. Results

### 3.1. Chemical Mutagenesis

For random chemical mutagenesis, *M. luteus* trpE16 ∆*comEA*/*EC*:*lacZ*-*Kan* reporter strain cells from an O/N full LB medium culture were used to inoculate LB_Kan_ with and without supplementation of 5 mM EMS. Mutagenesis efficiency was assessed by determining the frequency of rifampicin resistant mutants induced by the chemical mutagen. The EMS-treated samples revealed 2.8 × 10^3^ CFU/mL capable of growth on 1 µg/mL of rifampicin while non-treated controls of the reporter strain and the wild-type yielded an average of less than 3 CFU/mL. The total cell count of all assayed cultures was about 108 CFU/mL when grown without any antibiotics. Thus, the frequency of rifampicin resistant mutants in the EMS treated cells was 1000-fold higher than the non-treated cells (Appendix A).

### 3.2. Mutant Screening and Phenotype Selection

EMS-treated cells were screened for mutants with altered expression of the *comEA/EC* promoter, which regulates *lacZ* gene expression in our reporter strain in dependence of the growth medium (Figure 1A). For this, mutagenized cells from a glycerol stock were inoculated in full medium and incubated O/N at 30 °C before plating serial dilutions on different media. EMS mutations leading to an increased promoter activity under competence-repressing conditions were identified on LB_Kan_ agar supplemented with X-gal (M-mutants), on and those repressing promoter activity under competence-promoting conditions were identified on GMM_Kan_ agar supplemented with X-gal (R-mutants). Use of the chromogenic LacZ substrate X-gal enabled a blue/white plate assay for mutant screening instead of a MUG plate-spraying technique with fluorescence read-out [28]. When looking for M-mutants with the MUG plate-spraying assay, it was not easy to qualitatively distinguish colonies displaying upregulation of the reporter gene construct from regular colonies. The fluorescence was too homogenous when many colonies were close to each other, making it difficult to distinguish between colonies with variations in fluorescence intensity. This method was also inconvenient for the screening of the R-mutants, given its lower sensitivity. Conversely, with the use of X-gal at relatively high concentrations (80 µg/mL), it was possible to differentiate strongly repressed phenotypes from the background of colonies with unchanged reporter gene expression. Although the expression of *comEA/EC* is normally inhibited in full medium, M-mutants exhibited a clear and strong expression, yielding blue-colored colonies on LB_Kan_/X-gal (45 µg/mL) plates (Figure 1B). Conversely, R-mutants were selected as white colonies among a background of blue colonies on GMM_Kan_/X-gal plates. Such phenotypically white colonies indicated repression of the *comEA/EC* promoter under competence-inducing conditions. Thus, compared to the wild-type, both groups of mutants failed to show the usual response of the *comEA/EC* promoter to nutrient availability.

More than 100 colonies of M-mutants showing enhanced activity of the β-galactosidase marker under repressing conditions and around 80 colonies of R-mutants with no or reduced β-galactosidase activity under inducing conditions were repeatedly re-streaked in order to allow genetic segregation of the mutations and to find stable phenotypes.

In order to assess which mutants of each group to choose for genome sequencing, a quantitative β-galactosidase activity assay was performed. For the M-mutants, we identified which mutants had high levels of expression of the *lacZ* reporter gene on full medium. For the R-mutants, which were expected to have largely reduced or lacking β-galactosidase activity, we selected strains that had less than half of the activity of the unmutagenized *lacZ* reporter strain using an end-point measurement (t = 10 min). Besides that, we wanted to evaluate if there were cases of SNPs occurring within the 7 kb reporter construct inserted in the genome, meaning higher genetic linkage of the causative SNP and the promoter. Fourteen R-mutants and thirty-two M-mutants fulfilled both the reporter activity and the genetic linkage screening criteria (less than 3% of linage; for linkage scores see Appendix A), and from these, all the R-mutants and 22 of the M-mutants were grown to stationary phase for high quality gDNA preparation and sequencing.

### 3.3. Genome Sequencing and Mutation Analysis

Whole-genome sequencing of the mutants was conducted with at least 200-fold genome coverage. Read mapping and SNP calling were performed by the Snippy script. Snippy finds SNPs between a haploid reference genome (*M. luteus* trpE16 in this case) and the NGS sequencing reads. The obtained data were used to generate a table containing each SNP position, the original nucleotide, and the new variant present in each mutant. As expected, after using EMS as mutagen [38], almost all documented mutations were G:C→A:T transitions. There were on average nine SNPs per genome in the M-mutants and eight SNPs per genome in the R-mutants. Besides the positions and effect of the mutations, the affected loci and their annotations were also considered. By using these data, an interactive plot was created for each set of mutants, in which each mutant along with its mutation profile and the position of each individual SNP could be inspected (Figure 2 and Appendix A). It was possible to observe the general distribution of the mutations along the *M. luteus* genome and to analyze how many SNPs there were per nucleotide position or if an increased SNP density was observable in specific genes or regions along the chromosome. To systematically detect sequence regions with higher SNP densities, a programmable “sliding window” function was implemented for data analysis in which a sequence “window” with adjustable length (e.g., 5000 bp) screens a defined number of positions (e.g., every 500 bp) along the genome sequence, producing a score at each step. This score is a measure for the number of SNPs within the covered area and of the distance between them. The more SNPs within the window and the closer their positions, the higher the final score. In this way, two windows with the same number of SNPs will have different scores if the distance between the SNPs is different. Sixteen different clusters, i.e., regions with increased SNP incidence, were recognized for the M-mutants (Figure 2), while for the R-mutants there was not a clear clustering of the SNPs (Appendix A). Beside the SNPs belonging to specific clusters, all SNPs affecting genes for transcription factors, proteases, or having other annotations of interest were also considered to decide if they should be included in further investigation. The most interesting clusters are listed in Table 2 and Table 3. One region (cluster 13) in particular revealed a high density of SNPs (Figure 2). Almost every M-mutant was found to have an SNP in this region, and in addition, one R-mutant (R4) with a strongly repressed phenotype (Appendix A) carrying an SNP within the same area. All mutations in cluster 13 were within a region of 550 bp that encompasses two genes, Mlut_14660 encoding a DNA binding protein with a Helix-turn-Helix 17 motif domain and Mlut_14650 encoding a putative Flp pilus assembly protein RcpC/CpaB with an annotated SAF domain. The ORF for the latter protein forms an operon along with Mlut_14640 and Mlut_14670*,* which encodes the putative Flp pilus assembly ATPase CpaE and a two-component sensor histidine-kinase containing a PAS domain, respectively (all annotation information is from NCBI). The intergenic region between Mlut_14660 and Mlut_14650 was also affected by two different SNPs (see zoomed part in Figure 2). Among other affected genes, there is Mlut_09880 and Mlut_09900, which code for the Clp protease proteolytic and ATP-binding subunits, respectively. The only M-mutant which did not carry an SNP within cluster 13 had an SNP in Mlut_02150, which encodes a protein with the same annotation as the DNA binding protein encoded by Mlut_14660. Due to the lack of clear clustering in the R-mutants, certain mutations were selected after visual inspection of the affected genes, considering the number of SNPs affecting them.

### 3.4. Introduction of SNPs and Gene Deletions into M. luteus and Evaluation of comEA/EC Promoter Activity

In order to confirm that individual point mutations found in the genomes of M- and R-mutants are responsible for increased or, respectively, decreased *comEA/EC* promoter activity as indicated by the observed blue/white phenotypes, we introduced single conspicuous mutations into the genome of non-EMS-treated *M. luteus* trpE16. The markerless introduction of point mutations into the genome was performed using a *codB/codA*-based counter-selection system [34]. Because of the lack of transformability of the reporter strain *M. luteus* trpE16 *∆comEA/EC:lacZ-Kan*, which lacks functional *comEA* and *comEC* genes, the genome editing had to be conducted in the trpE16 parental strain. After confirmation of successful SNP re-insertion by sequencing, each single-SNP-containing strain was transformed with a DNA fragment carrying the *∆comEA/EC:lacZ-Kan* construct, leading to strains with single point mutations, which otherwise were isogenic to the original reporter strain *M. luteus* trpE16 *∆comEA/EC:lacZ-Kan*. This allowed the comparison of the LacZ reporter enzyme activity between isogenic strains, differing only by selected point mutations or gene deletions. In total, nine SNPs belonging to four different clusters were re-inserted successfully as single chromosomal point mutations, and three of the affected genes were cleanly knocked out (Table 3). Despite numerous attempts with the *codBA* counter-selection system and also by *kan* exchange through homologous recombination, it was not possible to isolate viable deletion or gene exchange mutants for Mlut_14660.

Once several SNPs of interest were individually re-introduced into the genome along with the *lacZ* reporter construct, the *comEA/EC* promoter expression levels of those strains were measured through the β-galactosidase/MUG activity assay. For these LacZ reporter assays, we focused on strains with single-SNP chromosomes constructed based on SNPs identified in M-mutants only. Each single-SNP mutant was compared with the original reporter strain *M. luteus* trpE16 *∆comEA/EC:lacZ-Kan* and evaluated along with its parental M-mutant and a knockout strain of the affected gene (if available) (Figure 3). From the nine single-SNP mutant strains assayed, three showed no difference compared with their respective parental M-mutant: the single-nucleotide mutants 13-M27, 13-M93, and 13-M1090 showed similar substrate conversion rates as the M-mutants M27, M93, and M1090, respectively (Figure 3). The Δ14650 knockout strain revealed a clear increase in activity of the *comEA/EC* promoter under competence-repressing growth conditions in LB medium, and this increase was significantly higher than the activity increase observed with the point mutations in the Mlut_14650 gene of strains M27 and 13-M27. All the other single SNP mutants (3-M43, 7-M28, 17-M33, 7-M89, 13-M89, and 7-M93) did not show a significant increase of *comEA/EC* promoter activity when compared with the reporter strain, indicating that these point mutations were not solely responsible for the observed blue/white phenotypes of the corresponding EMS mutants originally isolated via the mutant screening and phenotype selection procedure described above.

### 3.5. Transformation Frequency Evaluation

An important question was if the selected and individually reinserted single-SNP mutations had a direct impact on *M. luteus* transformability. To answer this question, transformation frequency assays were conducted. *M. luteus* trpE16-derived strains with edited single-SNP chromosomes or with clean gene deletions (see above) were cultured in GMM broth before cells were harvested and transformed with pJET-*trpE* plasmid to assess the frequency of conversion to Trp prototrophy. The mutant strain 13-M89 showed a 100-fold decrease in transformability compared to the parental strain *M. luteus* trpE16. Mutants *M. luteus* trpE16 2-R33 and *M. luteus* trpE16 8-R34 had lower frequencies of transformation, but not as low as the knockout *M. luteus* trpE16 ∆14650 or *M. luteus* trpE16 13-R4 (Figure 4). This last mutant is the only mutant that had a transformation frequency well below the detection limit of the assay. Consequently, it was not feasible to construct a ∆*comEA*/*EC*:*lacZ*-*Kan* reporter strain after introducing the corresponding point mutation (G1606821A) into *M. luteus* trpE16. The transformation frequencies of the *M. luteus* trpE16 mutants 13-M93, 13-M1090, 13-M27, ∆02150, and ∆03135 did not differ significantly from the reference strain *M. luteus* trpE16.

### 3.6. Genomic Synteny of Cluster 13-Associated Genes

As cluster 13 mutations showed a strong influence on the *comEA/EC* promoter, and certain mutations from this cluster affected the transformability of *M. luteus**,* we evaluated the presence, distribution, and genomic context of the genes in this cluster and flanking genes in other members of *Actinobacteria* using the microbial genomic context viewer. BLAST was used to obtain the E-values of the homologs with *M. luteus* gene sequences as queries. Similar gene arrangements as at cluster 13 of *M. luteus* were found in species of the genera *Geodermatophilus, Rothia*, *Jonesia*, and *Cellulomonas*. In *Nocardioides*, the genes were also present, although they had a different orientation relative to each other. For *Gordonia* and *Kytococcus* species, the analysis revealed that homologues of the genes are present, but in a different organization (Figure 5).

## 4. Discussion

### 4.1. Mutant Selection and Clustering of Mutations

The most fundamental and conserved proteins involved in natural transformation are *ComEC*, a polytopic integral membrane protein that forms the DNA uptake channel, and the extracytoplasmic DNA receptor *ComEA*. Their importance is also true for the actinobacterial model organism *Micrococcus luteus* [4,15]. We aimed to find out more about how these co-transcribed genes are regulated. Information about the genes involved in competence development for natural transformation and its regulation in the *Actinobacteria* is scarce. It was only recently shown that the pilus-like structure, which is commonly found to be necessary for natural transformation in different bacterial lineages (usually type IV pili or pseudo-pili, and in certain cases, type IV secretion system), is a Flp pilus structure in *M. luteus* and is thus different from other bacteria studied previously. Orthologs of most other genes implicated in competence development and natural transformation in the best-studied low-GC-Gram-positive and Gram-negative model bacteria were not found in *M. luteus* [15,26]. Therefore, we decided to take a classical random mutagenesis approach for the search for gene candidates involved in this bacterium. In the *Firmicutes* model bacterium *Bacillus subtilis*, Dubnau and others in the 1980s isolated 28 *Tn917-lacZ* mutants that were deficient in competence as well as mapped the mutations and characterized the mutants with respect to DNA binding and uptake [39,40]. The expression of β-galactosidase under the control of the *com* promoter was studied as a function of growth stage, medium, and genetic background [41]. Later, Dubnau and Roggiani [27] used EMS-induced chemical mutagenesis to generate mutants that would permit the overexpression of *com* genes under normally repressing conditions, such as in full medium. The *mec* (for medium-independent expression of competence) mutations found by this approach mainly affected two genes, *mecA* and *mecB.* Later, it was determined how they affected the regulation of medium-dependent expression of competence [42].

Our group previously constructed a *lacZ* reporter strain to study expression of the late *com* genes *comEA/EC* in *M. luteus* [26]. We chose EMS random chemical mutagenesis as an efficient method for introducing random mutations into *M. luteus* trpE16 *∆comEA/EC:lacZ-Kan*. On average, we obtained one stable transition every 3 × 10^5^ bp. If lower, the frequency of appearance of a desired phenotype would make the screening for its detection laborious, whereas if it were higher, it would be difficult to pinpoint the causative mutation for a specific trait among numerous other point mutations in the genome. Regarding the mutagenesis efficiency, our results are within the same order of magnitude as described in the study on *B. subtilis* by Dubnau and Roggiani (1990) [27].

It was crucial to filter out EMS mutants that exhibited the medium-dependent overexpression or repression phenotype of interest but were genetically closely linked to the *comEA/EC* locus or the *lacZ* reporter gene because it was our goal to identify new loci involved in competence development that are different from *comEA**/EC*. By natural transformation of the wild-type strain with gDNA isolated from the M or R mutants, it was possible to analyze the degree of linkage between the phenotype-causing SNPs and the seven kbp ∆*comEA*/*EC*:*lacZ*-*Kan* reporter construct region. A linkage analysis similar to the LOD score [43] was conducted to study *cis*-acting mutations in *comG* of *B. subtilis* [28]. In our study, the linkage analysis was successfully used to identify and focus on *M. luteus* trpE16 EMS mutants that were genetically unlinked to the *comEA/EC* locus.

The genomes of 22 M-mutants and 14 R-mutants were sequenced and variant calling was performed by using the Snippy program tool, which finds high confidence differences (indels or SNPs) between a known reference genome and sequencing reads [44]. Sixteen SNP clusters were identified for the group of M-mutants, whereas for the R-mutants, there was no clear clustering. There were almost no loci where mutations were found in more than one R-mutant. This could mean that there may be more than one region in the genome that by mutation can generate partial or total repression of *comEA/EC* promoter. It will be interesting to see if by increasing the EMS exposure time or the number of R-mutants sequenced and aligned against the reference genome, a clearer clustering of mutations connected with the phenotype of *comEA/EC* promoter repression under competence-inducing conditions shows up.

Of the 16 SNP clusters identified for the group of M-mutants, those with higher numbers of positions affected and those with a higher number of mutants carrying one of the mutations of the respective cluster were selected. Besides the regions with higher clustering scores, another selection criterium was the annotation of the genes affected by the mutations, prioritizing DNA-binding proteins, transcription factors, proteases, response regulators, or other genes judged to possibly participate in a competence regulatory network. The most interesting clusters were clusters 3, 7, and 13. Since, for the R-mutants, the mutation clustering was not as clear as with the M-mutations (Appendix A), the selection of mutations for further investigation was mainly based on visual analysis of gene annotations. Since each EMS mutant strain contained several point mutations, it was crucial to re-insert conspicuous mutations as single nucleotide exchange mutations into the genome of *M. luteus* trpE16 and to re-insert the 7 kbp ∆*comEA*/*EC*:*lacZ*-*Kan* reporter construct and determine the transformability of the single-SNP mutants in comparison to the *M. luteus* trpE16 wild-type. Three genes found to be affected by EMS mutations in this study were deleted in order to evaluate the resulting phenotype for further confirmation of the genes’ roles.

### 4.2. R-Mutations

Four positions were affected within cluster 2 of the R-mutants (Table 3). Two missense transitions within Mlut_07500 encoding the Flp pilus assembly protein/NTPase TadA1/CpaF and another one in Mlut_07560, a hypothetical Flp pilus assembly TadG-like protein. Both proteins belong to the previously described *tad-1* cluster and have been shown to be essential for transformation/pilus assembly [15].

Three R-mutations that were selected for re-insertion as single-nucleotide exchange mutations gave rise to strains (2-R33, 8-R34, 13-R4) with at least 1000-fold lower transformability than the wild-type in late-exponential phase grown in GMM (Figure 4). Mutation G1541663A/D149N (8-R34) affected Mlut_14110, an ORF encoding a putative response regulator that forms an operon along with the putative histidine kinase ORF Mlut_14100, together probably composing a two-component signal transduction cascade similar to *sivS/R* of *Streptococci iniae*, according to a BLAST search [45]. Mutation G1606821/G27D (13-R4) affects ORF Mlut_14650 and is merely 10 bp upstream of mutation G1606810A/V31I (13-M27) from cluster 13 of the M-mutants, which is discussed below.

Mutation G810889A/G327E (2-R33) affects Mlut_07490, an ORF encoding an AAA+ ATPase DNA translocase of the FtsK/SpoIIIE family, which is directly adjacent to the *tad-1* cluster, one of two tad gene clusters already known to be essential for natural transformation in *M. luteus* [15]. Members of this protein family are involved in a wide variety of processes in many species, such as conjugation, DNA packaging, DNA segregation, and maintenance [46]. Mlut_07490 seems to share structural features with FtsK and SpoIIIE, such as a well conserved DNA translocation module and transmembrane helices (four in FtsK and SpoIIIE, but only two in Mlut_07490; sequence alignment in www.uniprot.org/align (accessed on the 18 August 2021) Identifiers: P46889, P21458, and C5C9X9). These transmembrane domains are localized at the proteins’ N-termini, usually followed by a sequence of variable length that interacts with other proteins, and at the highly conserved C-terminus there is a DNA-binding module [47,48] whose specific role in *M. luteus* has not yet been studied. Mlut_07490 has the typical highly conserved P-Loop consensus amino acid sequence “GTTGSGKS” (Walker A motif). The Walker B motif is usually located downstream of Walker A and has a higher variability. The only invariant features of this motif are a negatively charged residue following a stretch of bulky, hydrophobic amino acids [49], such as the “AAAAPAVAPWE” present in *M. luteus* Mlut_07490. In a study conducted on *B. subtilis* [50], a *spoIIIE* mutation was the only one that significantly decreased the *lacZ* reporter gene expression (in that case under control of the *spoIIIG* promoter). This suggests that the *spoIIIE* product is directly involved in a mechanism of differential gene expression. The membrane-spanning and mononucleotide-binding domains suggest that the protein may function as part of a signal transduction pathway based on phosphoryl group transfer, similar to the NtrB/NtrC system in other bacteria [51].

When transforming the single-nucleotide exchange mutants 2-R33 and 8-R34 with the *lacZ* reporter construct, blue colonies were obtained. This indicates that although these SNPs affect transformability, they do not directly repress *comEA/EC* promoter expression, suggesting that they may play a role in the process at a different level or that one or more other mutations are required for *comEA/EC* promoter repression.

### 4.3. M-Mutations

Cluster 3 of the M-mutants encompasses six mutants with two positions affected within the same gene, Mlut_03135, which codes for a N6-adenine DNA methyltransferase of unknown recognition sequence. These type of enzymes are widely spread in prokaryotes, and besides their role in restriction-modification systems [52], they are an important method of regulation of gene expression. Changes in promoter methylation status has been shown to control several genes in different model organisms such as *E. coli* and *Caulobacter sp*. A transcriptomic study conducted in *E. coli* reported many deregulated genes in DNA adenosine methyltransferase mutants, including global regulators such as catabolite activator protein or fumarate nitrate reductase [53]. However, the single-nucleotide mutant derived from M43, 3-M43, and the corresponding gene deletion mutant trpE16 ∆Mlut_03135 did not show increased activity of the *comEA/EC* promoter, suggesting that promoter adenine methylation by the putative methylase encoded by Mlut_03135 probably does not play a role in regulation.

Cluster 7 of the M-mutants is composed of three positions affected in six different mutants. The affected loci, Mlut_09880 and Mlut_09900, encode the AAA+ ATP-dependent Clp protease proteolytic subunit ClpP and the ATP-binding subunit ClpX, respectively. ClpX recognizes unstructured peptide tag sequences in protein substrates, proceeds to unfold stable tertiary structure in the protein, and then spools or translocates the unfolded polypeptide chain into a sequestered proteolytic compartment in ClpP for degradation into small peptide fragments [54]. The levels of the main transcriptional regulator ComK of *B. subtilis* are controlled by this type of proteolytic system, composed of the adapter protein MecA (see Introduction) and the ATP-binding subunit ClpC (formerly known as MecB) [27,Mlut_0749042]. In *B. subtilis*, mutants in *mecA*, ClpC, and ClpP lead to the overexpression of *com* genes and other genes activated by ComK. This scenario may be similar to what was observed in the M-mutants carrying changes in the region of cluster 7.

However, none of the mutations from cluster 7 (i.e., strains 7-M28, 7-M89, and 7-M93) restored the screened EMS mutants’ *comEA/EC* promoter activity when introduced into the *M. luteus* chromosome as single-SNP mutations (Figure 3). This means that at least the effect of those single mutations on the Clp proteases are not enough to affect *comEA/EC* promoter expression rates. None of the aforementioned single-SNP M-mutants or gene knockout mutants had significant changes in their transformation frequencies in comparison to the trpE16 wild-type strain in late exponential phase GMM broth cultures. It is possible that the resulting phenotypes for certain M- or R-mutants depend on the combined effect of more than one mutation. Further genome editing experiments are needed to see if the combination of certain single-SNP mutations, such as those from strains 7-M89 or 13-M89 which have an unaltered *comEA/EC* expression phenotype compared with the non-mutagenized ∆*comEA*/*EC*:*lacZ* control strain (Figure 3), resulting in the same LacZ reporter phenotype as their parental strain M89.

Cluster 13 encompasses six different positions, with one of them detected in almost every M-mutant, the only exception being M33. Thus, the highest density of SNPs identified by our phenotypic screening (see Figure 2) was in this region of the genome. ORFs of two apparently divergently overlapping transcriptional units were affected: G1606810A/V31I (13-M27) lies within the Mlut_14650 ORF, which codes for a SAF-domain containing protein with similarity to RcpC/CpaB Flp pilus assembly proteins, while G1607081A (13-M89) and G1607117A (13-M1090) putatively affect the 5′UTR of Mlut_14660, and mutations C1607204T/S27L (13-M96), C1607261T/S46F (13-M9) and C1607314T/R64W (13-M93) affect the Mlut_14660 ORF. The only R-mutation identified in this cluster was 13-R4, which has the mutation G1606821A/G27D in the Mlut_14650 ORF.

The mutations affecting Mlut_14650 showed different phenotypes. 13-R4 showed the strongest phenotype with a more than 10^5^-fold decrease in transformation frequency in comparison to the wild-type, which is below the detection limit of the assay. The amino acid changed in the protein encoded by Mlut_14650 of strain 13-R4 (G27D) is only four residues apart from the missense mutation (V31I) present in the same protein of strain 13-M27, which did not show a change in transformability (Figure 4). In agreement with the drastic transformation phenotype observed in 13-R4, a clean knockout of Mlut_14650, obtained using the *codBA* deletion system, resulted in a transformability loss of 10^4^-fold for this mutant, indicating a significant role of Mlut_14650 in the regulation of natural transformation.

Among the M-mutants affecting the 5′UTR of Mlut_14660, 13-M89 revealed a decrease of between 100- and 1000-fold in transformability while 13-M1090 did not show a transformation phenotype. According to what was predicted by several mRNA secondary structure prediction programs, the transition of 13-M89 affects a highly conserved loop that may be necessary for correct translation of the encoded DNA binding protein (Figure 6). Evidence of the role of 5′ UTR folding on translation initiation and in post-transcriptional regulation was found [55]. The single-nucleotide mutation in 13-M1090 does not seem to change any secondary structure according to the prediction programs used. Instead, it modifies the Shine-Dalgarno sequence and thus may affect ribosome binding and translation initiation rates (Appendix A). The missense mutation C1607314T in mutant 13-M93 generates a R64W exchange in the Mlut_14660-encoded putative DNA binding protein. M93 was the only mutant to hold this particular mutation, and after its insertion in the reporter strain, the phenotype of the parental mutant strain M93 was fully regained (Figure 3).

The small Mlut_14660 gene encodes a DNA-binding protein with an HTH-17 domain annotated as a member of the excisionase protein family. It was not possible to obtain a Mlut_14660 knockout strain after having tried with different techniques such as *codBA* clean deletion or *kan* exchange, and by using different lengths of flanking homologous DNA in the deletion and replacement constructs. It is therefore possible that the protein encoded by this gene may be essential for *M. luteus* and may be involved in the regulation of more than one cellular process such as natural competence or transformation. Conversely, the gene Mlut_02150 was knocked out without trouble, indicating that although they have similar annotations, they are not interchangeable or have the same cellular task. The operon upstream of and divergently oriented to Mlut_14660 encompasses the three structural genes Mlut_14650, Mlut_14640, and Mlut_14670, coding for the putative Flp pilus assembly protein RcpC/CpaB mentioned above, a TadZ/CpaE pilus assembly protein, and a PAS domain-containing sensor histidine kinase, respectively. The RcpC/CpaB homologue in *M. luteus* possesses an N-terminal signal peptide (predicted by SignalP 5.0 software) but no internal transmembrane domains, suggesting that the protein is located outside the membrane, such as its homologues in other organisms [56]. RcpC, encoded in the prototypical *tad* locus of *Aggregatibacter actinomycetemcomitans*, and CpaB in *Caulobacter crescentus* and all its known homologues contain two β-clip motifs involved in a variety of functions, including binding carbohydrate moieties to assemble structures such as Flp pili [46,57]. It has also been proposed that one or both β-clip motifs glycosylate pilin subunits to facilitate their assembly. As indicated by Tomich et al. [56], a rcpC/cpaB gene is absent in the tad loci of Gram-positive bacteria, which also holds true for *M. luteus* [15]. This does not indicate that such genes cannot still be present at a different genomic locus and belong to a different transcriptional unit. The same applies for *tadZ/cpaE*, although this gene is indeed present in *tad* clusters of other actinobacteria [56]. This cytosolic soluble protein is an essential component of the Tad secretion system in many organisms, and its particular molecular architecture combines components from the bacterial cytoskeleton (MinD/ParA/Soj ATPases) and two-component signal transduction response regulators. Studies in *A. actinomycetemcomitans* and *C. crescentus* showed that they probably function as localization factors [58,59]. In *C. crescentus*, CpaE mediates positioning of the pilus secretin protein CpaC, as well as the histidine kinase PleC. Proteins containing PAS-domains such as the one encoded by Mlut_14670 are usually modular, functioning as sensors that detect a variety of stimuli and regulate, in response, the activity of effector (output) domains with functions such as catalysis or DNA binding [60]. In certain organisms, such as *P. aeruginosa* [61], *A. actinomycetemcomitans*, *A. aphrophilus* [57] and *C. crescentus* [62,63], the environment-dependent modulation of pilin production through two-component regulatory systems of this type seems to be a common feature.

Considering how close together Mlut_14660 and Mlut_14650 are, it is difficult to assess whether mutations in the 5′ regions of these genes, which also may encompass their regulatory sequences, affect one gene or the other, or both, and in which way. To obtain more information on this matter, the starting point of each transcript unit was determined. It seems that there is a divergent overlap of the transcriptional units and 5′UTRs (97 bp apart) of Mlut_14660 and the Mlut_14650/*40*/*30* operon structure, including the promoter or operator sequences of each unit and maybe extending into part of the coding regions. This type of arrangement [64] benefit the hypothesis of transcriptional interference (TI) as a way of gene regulation of Mlut_14660 and the operon Mlut_14650/40/30. TI was defined as the suppressive influence of one transcriptional process, directly and in cis on a second transcriptional process by Shearwin [65]. A model such as this one was also proposed for the divergently overlapping *comEA/EC* and *comER* genes in *B. subtilis* [16]. In a more recent work [66], it has been shown that *comER* plays an important role in biofilm formation and sporulation, and that competence has a negative correlation with biofilm formation [67], supporting the stated hypothesis. When analyzing if the gene Mlut_14660 and the operon Mlut_14650/40/30 and their specific genetic context is conserved in other bacteria, we found that they are present and have the same or similar surroundings on the chromosomes of several different Actinobacteria genera (see Figure 5; only selection of chromosomal regions of related genera and species with similar gene arrangements depicted). In the proposed model (Figure 7), the expression of one of the transcripts may generate a negative superhelical turn in the opposite sense, impeding the expression of the other transcript. The steric hindrance derived from the RNA polymerase bound to the promoter can also play a role, or a mix of both phenomena. Data published recently showed that the Mlut_14650/40/30 operon is among the 25 most highly expressed *M. luteus* genes in mid to late exponential phase when grown in GMM. This is not the case when the same strain (trpE16) was cultured in LB [68]. The direct implication of Mlut_14650/40/30 in pili formation still needs to be verified. Further work is needed to verify the binding of the Mlut_14660 product to the *comEA/EC* promoter.

## 5. Conclusions

In conclusion, random EMS mutagenesis followed by introduction of specific mutations into the *M. luteus* genome as single-SNP mutations and markerless gene deletions has allowed the identification of new genes previously not reported to be involved in competence development and natural transformation. While certain constructed single-SNP mutations did not affect *com* gene expression and natural transformation on their own and possibly only exert an effect when combined with other SNPs, single-SNP mutations in Mlut_14660 and in the 5′ regions of Mlut_14660 and Mlut_14650/40/30 clearly affected *com* gene expression and natural transformation. Further work is planned to confirm the hypothetical regulation model for these genes presented here and to unravel the molecular basis of competence regulation in the Actinobacteria model organism *M. luteus*.

## Figures and Tables

**Figure 1 genes-12-01307-f001:**
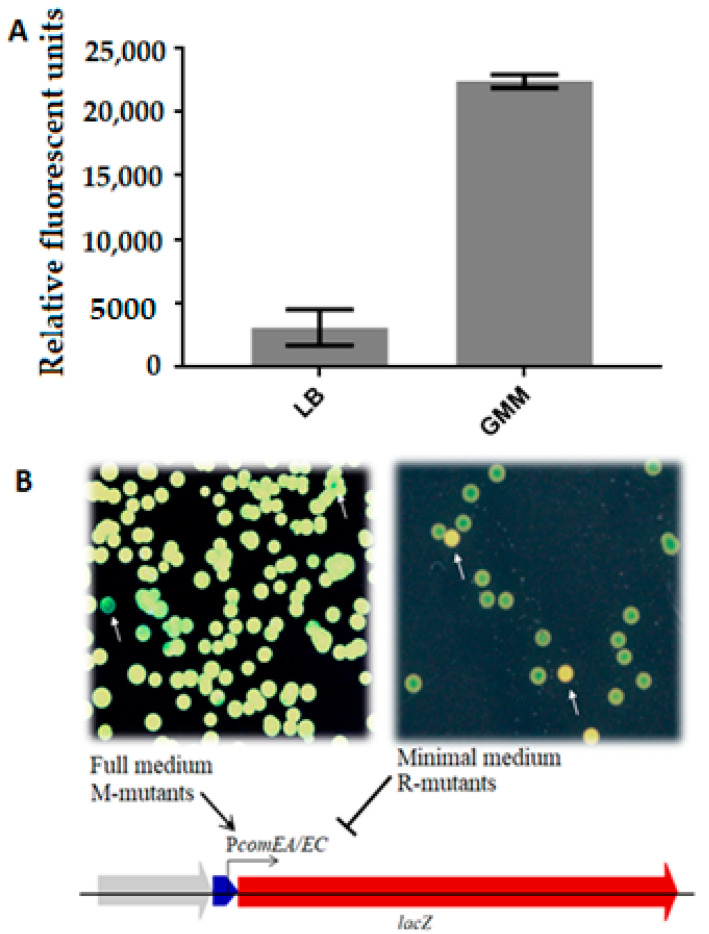
*M. luteus**comEA/EC* promoter activity and screening for EMS mutants with dysregulated activity on rich and minimal medium. (**A**) End-point measurement (t = 30 min) of *comEA/EC* promoter transcriptional levels after growth of the reporter strains *M. luteus* trpE16 *ΔcomEA/EC:lacZ* under different nutritional conditions (lysogeny broth (LB) and glutamate minimal medium (GMM)) using the β-galactosidase substrate 4-methylumbelliferyl β-D-galactopyranoside (MUG). (**B**) Screening of EMS mutants with deregulated *comEA/EC* promoter. On the left panel, mutants (blue) show higher enzymatic activity, indicating an increase in the *comEA/EC* promoter activity under usually competence-repressing conditions (LB^Kan^ full medium with 45 µg/mL of 5-bromo-4-chloro-3-indolyl-β-D-galactopyranoside (X-gal)). The mutants with this phenotype were called M-mutants. The right panel shows colonies derived from EMS-mutagenized cells plated on competence-inducing minimal media (GMM^Kan^ with 80 µg/mL of X-gal), including mutants with no enzymatic activity (yellow due to natural color of *M. luteus* cells), which were designated as R-mutants. Arrow: Blue: *comEA/EC* promoter sequence; Red: *lacZ* gene sequence. Normal arrow indicates promoter induction; Blunt arrow indicates promoter repression.

**Figure 2 genes-12-01307-f002:**
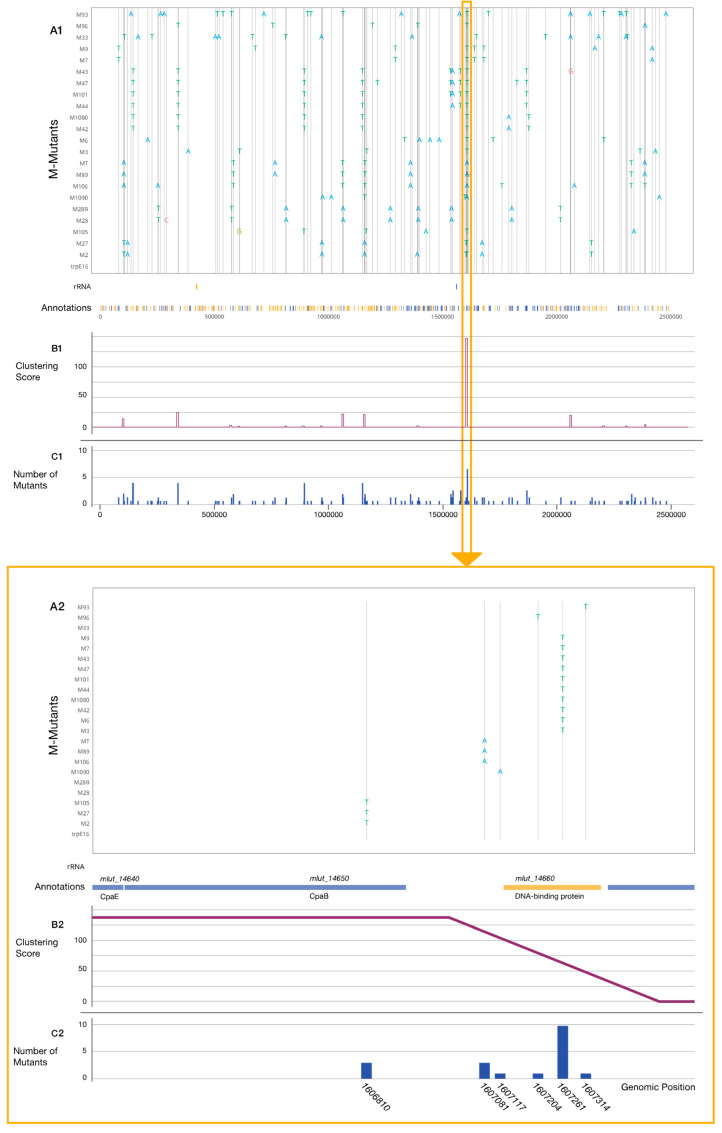
Distribution and clustering analysis of single nucleotide polymorphisms (SNPs) in M-mutants. All plots in the figure share the horizontal axis, which represents the reference genome of *M. luteus* trpE16. Every SNP of each mutant is shown in the upper panel and a zoomed view of a section thereof in the lower panel (framed in orange). In both panels: (**A1**,**A2**) all aligned M-mutants and their SNPs, along with the annotated genes affected by each mutation; (**B1**,**B2**) clustering score for each position, indicating the most affected areas in the genome. This score is the result of a programmable sliding window function that was applied throughout the genome. The window coverage was set to 5 kbp; (**C1**,**C2**) number of mutants holding a mutation at each position in the genome. The lower panel represents the enlarged area with the highest clustering score. Six mutated positions from the M-mutants belonging to cluster 13 are shown here. Directly under the SNPs plot, the affected genes Mlut_14650 and Mlut_14660 are drawn in blue and orange, respectively, according to their orientation on the DNA. Two mutated positions fall into the intergenic area, more specifically within the 5′UTR of Mlut_14660.

**Figure 3 genes-12-01307-f003:**
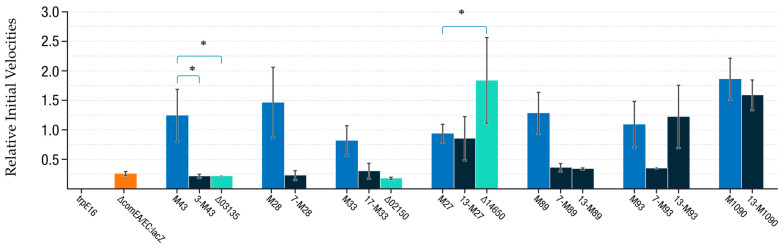
Relative initial velocities of β-galactosidase activity of *M. luteus* trpE16*, the ΔcomEA/EC:lacZ* reporter strain and its derivative mutants. *ComEA/EC* promoter transcriptional activities for *M. luteus* trpE16, the *ΔcomEA/EC:lacZ* reporter strain (orange), and its derivative mutants under normally competence-repressing conditions (LB full medium). All cultures were normalized to cell density prior to lysozyme treatment. LacZ substrate MUG was added and kinetic data for the promoter activity was recorded over 8 hours by measuring the fluorescence generated from the substrate cleavage. A four parameter logistic model was fitted to the kinetic curves (*n* ≥ 3). The slope of the logistic curve served as a measure for the relative promoter activity. Expression levels of the M-mutants are shown in blue, with the single SNP mutants in dark blue and the gene knockouts in turquoise. The mean values of ≥3 independent experiments and standard deviations of all experiments are presented. An unpaired, two-tailed Student’s t-test showed no difference between M27, M93, and M1090 and 13-M27, 13-M93, and 13-M1090, respectively. *, indicates statistically significant difference.

**Figure 4 genes-12-01307-f004:**
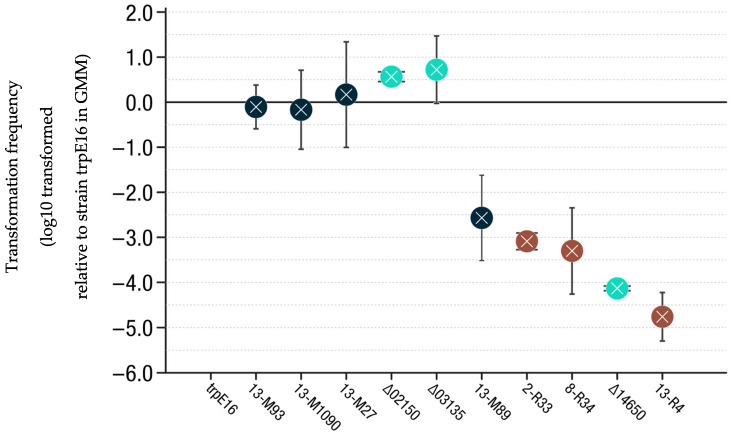
Effect of specific genome modifications on the transformability of *M. luteus*. Transformation frequencies of clean deletion mutants are shown in turquois, while single-SNP mutants with SNPs from the M-mutants and from the R-mutants are shown in dark blue and brown, respectively. Each dot represents the log_10_-transformed mean transformation frequency relative to the mean of the trpE16 strain. All strains were grown in GMM for 20 h until OD_600nm_ reached 1. The error bars indicate the SD of the biological replicates. An ordinary one-way ANOVA test was performed to compare all means with each other (*p* < 0.005, *n* ≥ 3).

**Figure 5 genes-12-01307-f005:**
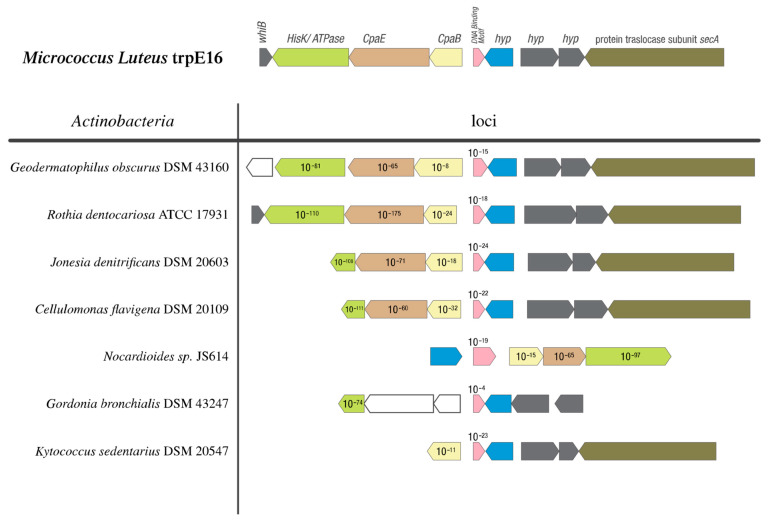
Physical maps of the Mlut_14660 and Mlut_14650/40/30 genes and their genomic surroundings on the chromosomes of a selection of actinobacterial (high-GC Gram-positive) representatives and from model organism *M. luteus*. ORFs with the same color encode proteins with significant similarity (BLAST E < 10^−4^) to the respective gene products of *M. luteus*. The predicted gene products are shown and the numbers in the arrows representing homologs are E-values of a BLAST search using the full-length *M. luteus* genes as queries. Pink: Mlut_14660; light yellow: Mlut_14650; light brown: Mlut_14640; green: Mlut_14630; blue: Mlut_14670. All representatives harbor the genes comEA and comEC in the chromosome.

**Figure 6 genes-12-01307-f006:**
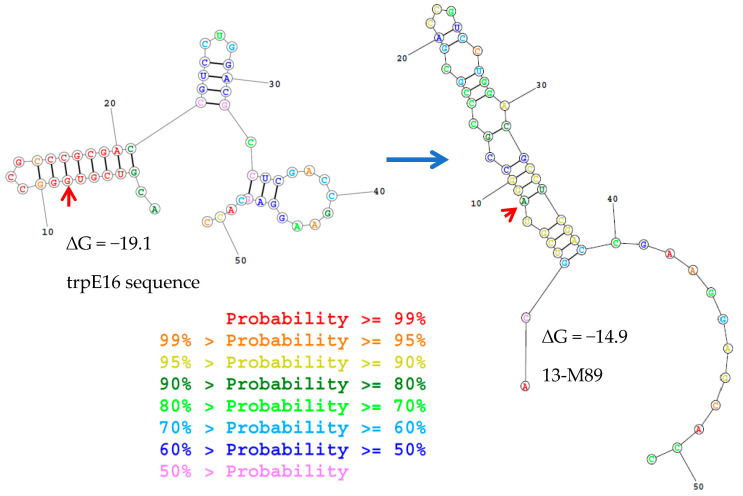
Output of secondary structure prediction program for the Mlut_14660 transcript. Most probable structures predicted through RNA secondary structure prediction tool, https://rna.urmc.rochester.edu/RNAstructureWeb/Servers/AllSub/AllSub.html (accessed on the 18 August 2021). Three loops can be observed for the original nucleotide sequence, with the one at the beginning of the transcript having the highest probability of formation. When the SNP 13-M89 is inserted, the previous structure is lost and another one forms.

**Figure 7 genes-12-01307-f007:**
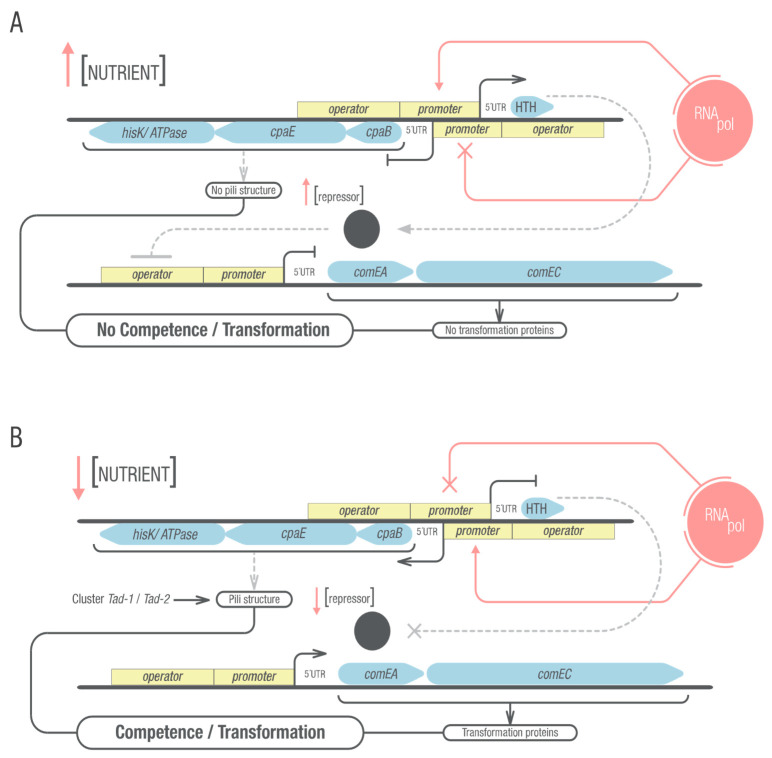
Transcriptional regulation model proposed for Mlut_14660 and Mlut_14650/40/30. As a working model, transcriptional interference is proposed for the divergently overlapping transcriptional units shown above. Grey dashed lines represent presumed but yet unproven effects. (**A**) Under non-competence-inducing conditions such as high nutrient concentrations, Mlut_14660 is expressed, impeding the expression of the Mlut_14650/40/30 operon which could play a role in pili structure building. The Mlut_14660 expression product, a putative DNA-binding protein with HTH motif, may cause the repression of essential *com* genes. (**B**) Opposite to the first scenario, under competence-inducing conditions such as low nutrient availability, the Mlut_14650/40/30 operon is expressed while simultaneously Mlut_14660 is not transcribed. This enhances *com* gene expression, resulting in higher abundance of transformation proteins and pili structure both necessary for DNA uptake. This model is based on results of this work and previous work but has not been verified in detail.

**Table 1 genes-12-01307-t001:** List of *M. Luteus* Strains Constructed in This Study.

Name	Genotype and Relevant Phenotype	Source
trpE16	*trpE16,* mutagenesis derivative of ATCC 27141, Trp^−^	[29]
Δ*comEA/EC*:*lacZ*	*trpE16* Δ*comEA*/*EC*:*lacZ*-*Kan*;Kan^R^, LacZ expression	[26]
∆14650	*trpE16 ∆*Mlut_14650	This study
∆02150	*trpE16 ∆*Mlut_02150	This study
∆03135	*trpE16 ∆*Mlut_03135	This study
3-M43	*trpE16* G341241A (Mlut_03135)	This study
7-M89	*trpE16 C*1061676T (Mlut_09880)	This study
7-M93	*trpE16* C1063999T (Mlut_09900)	This study
7-M28	*trpE16* G1064341A (Mlut_09900)	This study
17-M33	*trpE16* C226684T (Mlut_02150)	This study
13-M27	*trpE16* G1606810A (Mlut_14650)	This study
13-M89	*trpE16* G1607081A (Mlut_14660)	This study
13-M1090	*trpE16* G1607117A (Mlut_14660)	This study
13-M93	*trpE16* C1607314T (Mlut_14660)	This study
2-R33	*trpE16* G810889A (Mlut_07490)	This study
13-R4	*trpE16* G1606821A (Mlut_14650)	This study
8-R34	*trpE16* G1541663A (Mlut_14110)	This study

**Table 2 genes-12-01307-t002:** M-Mutants Main Mutation Clusters, Single Nucleotide Polymorphism (SNP) Features and Affected Genes.

Cluster	SNP	Mutant/s	Mut_Type	Locus	ORF Orientation	Annotated Product
1	G103107A	M89, M106, MT	Transition	Mlut_00960	>	Cof subfamily of IIB subfamily of haloacid dehalogenase superfamily
1	G103500A	M27, M2	Transition	Mlut_00970	<	Protein of unknown function DUF88
1	G106197A	M33	Transition	Mlut_00990	<	Amino acid transporter
3	G341241A	M43, M47, M101, M1080, M42	Transition	Mlut_03135	<	N6-DNA Methylase
3	G341448A	M96	Transition	Mlut_03135	<	Hypothetical protein
4	C575082T	M93	Transition	-	-	Intragenic sequence
4	G575822A	M28, M28B	Transition	Mlut_05280	<	Hypothetical protein
7	C1061676T	M89, M106, MT	Transition	Mlut_09880	>	ATP-dependent Clp protease proteolytic subunit ClpP
7	C1063999T	M93	Transition	Mlut_09900	>	ATP-dependent Clp protease ATP-binding subunit ClpX
7	G1064341A	M28, M28B	Transition	Mlut_09900	>	ATP-dependent Clp protease ATP-binding subunit ClpX
13	G1606810A	M105, M27, M2	Transition	Mlut_14650	<	SAF domain-containing protein
13	G1607081A	M89, M106, MT	Transition	-	-	Intergenic sequence (Mlut_14660 promoter)
13	G1607117A	M1090	Transition	-	-	Intergenic sequence (Mlut_14660 promoter)
13	C1607204T	M96	Transition	Mlut_14660	>	DNA-binding protein, excisionase family
13	C1607261T	M9, M7, M43, M47, M101, M44, M1080, M42, M6, M3	Transition	Mlut_14660	>	DNA-binding protein, excisionase family
13	C1607314T	M93	Transition	Mlut_14660	>	DNA-binding protein, excisionase family
15	C2060795T	M93	Transition	Mlut_19080	<	Flavodoxin
15	C2061073T	M33	Transition	Mlut_19090	<	Thiol-disulfide isomerase such as thioredoxin
15	T2061826C	M43	Transition	Mlut_19090	<	Thiol-disulfide isomerase such as thioredoxin
17	C226684T	M33	Transition	Mlut_02150	>	DNA-binding protein, excisionase family
18	G1878213A	M42, M1080	Transition	Mlut_17340	<	RNA polymerase sigma factor, sigma-70 family

**Table 3 genes-12-01307-t003:** R-Mutants Main Mutation Clusters, SNPs Features and Affected Genes.

Cluster	SNP	Mutant/s	Locus	Orientation	Annotated Product
1	C9672T	R42	Mlut_00060	>	DNA gyrase subunit A
1	G10325A	RE	Mlut_00060	>	DNA gyrase subunit A
2	G810889A	R33	Mlut_07490	<	DNA segregation ATPase, FtsK/SpoIIIE family
2	G812767A	R13	Mlut_07500	>	Flp pilus assembly protein, ATPase CpaF
2	C812940T	R16	Mlut_07500	>	Flp pilus assembly protein, ATPase CpaF
2	G816694A	R34	Mlut_07560	>	hypothetical protein
3	C934586T	R21	Mlut_08640	>	glutamyl-tRNA synthetase
3	G935940A	R16	Mlut_08660	>	tRNA-Gln
4	G1011921A	R4	Mlut_09340	<	2-oxo-acid dehydrogenase E1 component, homodimeric type
4	C1014740T	R13	Mlut_09380	>	4-azaleucine resistance probable transporter AzlC
5	G1029395A	R7	Mlut_09550	<	peptide deformylase
5	G1031740A	R13	Mlut_09580	>	translocating P-type ATPase, Cd/Co/Hg/Pb/Zn-transporting
6	G1249315A	R42	Mlut_11600	<	phosphoserine phosphatase SerB
6	C1250564T	R42	Mlut_11610	>	ABC-type molybdenum transport system
7	G1460599A	RE	Mlut_13330	>	2-oxoglutarate dehydrogenase E2 component
7	C1462336	R4	Mlut_13350	<	protein kinase family protein
8	G1541663A	R34	Mlut_14110	>	response regulator of citrate/malate metabolism
8	C1542830T	R41	Mlut_14120	<	acyl-CoA synthetase/AMP-acid ligase
9	G1814683A	R3, R30	-		Intergenic sequence, Mlut_16600 promoter
10	C1962653T	R4	Mlut_18140	>	helicase family protein with metal-binding cysteine cluster
10	C1964134T	R13	Mlut_18140	>	helicase family protein with metal-binding cysteine cluster
11	C2027721T	R3, R30	Mlut_18760	>	tRNA (guanine-N(7)-)-methyltransferase
11	C2028876T	R34	Mlut_18770	<	α/β hydrolase of unknown function (DUF1023)
12	G2427638A	R3, R30	Mlut_22740	<	hypothetical protein
13	G1606821A	R4	Mlut_14650	<	SAF domain-containing protein

## Data Availability

Not applicable.

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
