# Peer review of "Identification of New Chromosomal Loci Involved in com Genes Expression and Natural Transformation in the Actinobacterial Model Organism Micrococcus luteus"

_genes, 2021, doi:10.3390/genes12091307_

Round 1

Reviewer 1 Report

This manuscript entitled “Identification of new chromosomal loci involved in com genes expression and natural transformation in the actinobacterial model organism Micrococcus luteus” is genome wide mutational analysis as the authors successfully demonstrated that new genetic elements involved in regulation of the comEA-comEC competence operon in M. luteus. Moreover, authors showed the convincing experiment to validate the observation. This manuscript may encourage the researchers working in the field of bacteria to develop competence regulation. Below are my comments:

 Comments:

  1. Author should show the effect of ethyl-methanesulfonate (EMS) mutagen on comEA/EC promoter and LacZ expression.
  2. Author should pull down promoter through 5’-biotinylating to identify novel proteins bound with R-mutant and M-mutant.
  3. All bar graphs should be replaced with graphs that explicitly show 3 (or more) data points with mean and SD values (preferred over SEM).

Author Response

Dear reviewer,

Thank you very much for your comments and suggestions. Here are the responses to them:

Comment N°1: Effect of EMS on comEA/EC promoter or lacZ ORF.

The effect of EMS is generally the same throughout the DNA double-strand, creating on average one transition every 3xE^5 bp. Of course, this can potentially alter the comEA/EC promoter or lacZ ORF, but this effect is not relevant to our results, given that we have filtered out all mutants that encompass a causative SNP inside that region.

Comment N°2: Pull-down assay.

We are including this methodology in the follow-up project.

Comment N°3: bar graphs.

We are not really sure how to change the graphs. The purpose of the bar plots is to show the difference or not between different mutants, don´t you think that the bar plots are a more clear and strong way of showing/observing this than many individual points? For figure 3, our program takes all measurements from all repetitions and calculates the individual initial velocity for each repetition and the average (with SD) for each mutant, all this is available in an excel file on demand.

Reviewer 2 Report

Your research is well done; however there are still some minor concerns, which needs to be addressed and needs minor revision/explanation. I see great efforts in presenting the research problem and every step in the methodology. Despite this, I encountered a few ambiguities and noticed editing errors. Sometimes I marked them as below, for example citation errors, others similar, if they are, you have to verify and improve, because I suspect they are present in the rest of the paper.

Please check the use of strain description. Make sure if M. luteus wild type is called trpE16 or not.  For exemple: in the lane 204.

Lane 31: „…there was…”?

Figure S3: write a legend for the strains on the x axis, as above, and unify with fig S4.

Lane 134: Table 1: Just a proposition, give a citation in the brackets in the first rows of tables (for Kloos and Lichev).

Lane 136: Please, explain in growth conditions section the full medium composition or it was just a LB medium? If yes, give a note there to exclude any doubts.

Lane 138: Please verify the [27] as apropiate citation for M. luteus trpE16. Who constructed this reporter strain? Did you do?

Lane 140: „…..30 space °C”.

Lane 150: how many M and R colonies did you obtain? Did ypu choose one? And then used in following steps?

Lane: 153-160 and Fig. 1A. I am confused, in which  medium LB (full) or GMM (minimal) the M/R mutants had enhanced/no promoter activity. What the bars present?

Lane 158: please give [??] to unify the citation..

Lane 181: „….com promoter-lacZ reporter construct,…” give here Fig S1.

Lane 214 change x to × and check the rest along the manuscript.

Lane 236: Wrong citation? [34] instead of [35]?

Lane 252-253: „Prior to counterselection via the codBA marker, transformants were first selected on LB plates containing antibiotic.” What parameters for this selections? Give more details.

Lane 266-281: Please, give readers more details and the purpouse of this section.

Lane 283: Use your abbreviation O/N for overnight culturing.

Lane 304 „an overnight (O/N)” this abbreviation was previously explained in the lane 136. Correct along the manuscript.

Lane 304: please give more details about full medium?

Lane 307 „2.8 x 103 CFU/mL” change 2.8 × 103 CFU/mL  and verife along the manuscript.

Figure S1. Did you fuze it? Or other authors? Please cite if necessary.

Lane 308: wildtype or wild type or wild-type? In the fig. S3 you wrote wild type. Unify along the paper.

Lane 345-346: „For the M-mutants we identified which mutants had high levels of expression of the lacZ reporter gene on full medium”. What full medium?

Supplementary Material: arrange the contents in the order they are cited in the text. Figures before Tables?

Lane 352: In the Table S2. R-mutants linkage assay results it can be seen 20 of R mutants? Which of the fourteen mentioned here mutants are described in the table S2?

Lane 360: How many NGS reads, that is M/R mutants were compared to trpE16 strain? 32 of M mutants and 14 of R mutants? or 22 of Mutants and 14 of R mutants (as mentioned in the Lane 354)?

Figure 5: Write M. luteus in italics (3x)

Lane 514: Is there any other way to induce overexpression of com genes than classical random mutagenesis?

Lane 520: Please check all references cited in your paper. I see many mistakes. Readers should not guess which paper the authors really referred to.

Lane 527: I suspect that you made a mistake. It was [26] instead of [27]?

Lane 534: Dubnau and Roggiani is [27]?

Lane 563: I cannot view the Fig S5.

Author Response

Dear reviewer,

Thank you very much for your time and suggestions. Here are some answers to your comments:

Figure S3: write a legend for the strains on the x axis, as above, and unify with fig S4. Given that in figure S3 we are showing CFU/mL and in figure S4 the frequency of a traits appearance, it would be difficult to mix the graphs. This shows more clearly the crude results and then the calculated outcome from them.

Lane 136: Please, explain in growth conditions section the full medium composition or it was just a LB medium? If yes, give a note there to exclude any doubts. Note given.

Lane 138: Please verify the [27] as apropiate citation for M. luteus trpE16. Who constructed this reporter strain? Did you do? It was constructed by our gruoup for previous work and publication (Lichev, 2019 -> [26])

Lane 150: how many M and R colonies did you obtain? Did you choose one? And then used in following steps? I obtained/isolated around 100 colonies of M-mutants and 80 colonies of R-mutants, from those many lost their phenotype after successive re-streaking.

Lane: 153-160 and Fig. 1A. I am confused, in which  medium LB (full) or GMM (minimal) the M/R mutants had enhanced/no promoter activity. What the bars present?  LB/full medium is normally repressive, and GMM/minimal medium is normally inducing (this is shown in bar plot 1A. But the M-mutants are overexpressing comEA/EC promoter in full medium, and the R(Repressed)-mutants do not express it in GMM (this is what is being shown in figure 1B).

Lane 360: How many NGS reads, that is M/R mutants were compared to trpE16 strain? 32 of M mutants and 14 of R mutants? or 22 of Mutants and 14 of R mutants (as mentioned in the Lane 354)? As stated in lane 354, 14 R and 22 M-mutants were aligned against trpE16 strain reference genome.

Lane 514: Is there any other way to induce overexpression of com genes than classical random mutagenesis? There are other techniques, like promoter exchange or heterologous expression, that could be used for the specific purpose of overexpressing comEA/EC, although in our particular approach, we needed to change the sequences of other genes that in the end would generate that phenotype. This lets us link those genes with comEA/EC.

Lane 563: I cannot view the Fig S5. What is it exactly that you cannot see?. The main purpose of this figure is to show that there is no clustering as there was for the mutations of the M-mutants.

All the other corrections were introduced into the text.